# Anomalous behavior of critical current in a superconducting film triggered by DC plus terahertz current

Fumiya Sekiguchi [1] ✉, Hideki Narita [1], Hideki Hirori [1], Teruo Ono [1] & Yoshihiko Kanemitsu [1] ✉

The critical current in a superconductor (SC) determines the performance of many SC devices, including SC diodes which have attracted recent attention. Hitherto, studies of SC diodes are limited in the DC-field measurements, and their performance under a high-frequency current remains unexplored. Here, we conduct the first investigation on the interaction between the DC and terahertz (THz) current in a SC artificial superlattice. We found that the DC critical current is sensitively modified by THz pulse excitations in a nontrivial manner. In particular, at low-frequency THz excitations below the SC gap, the critical current becomes sensitive to the THz-field polarization direction. Furthermore, we observed anomalous behavior in which a supercurrent flows with an amplitude larger than the modified critical current. Assuming that vortex depinning determines the critical current, we show that the THz-current-driven vortex dynamics reproduce the observed behavior. While the delicate nonreciprocity in the critical current is obscured by the THz pulse excitations, the interplay between the DC and THz current causes a non-monotonic SC/normal-state switching with current amplitude, which can pave a pathway to developing SC devices with novel functionalities.

The most important feature of a superconductor (SC) is, in addition to the Meissner effect, a supercurrent that flows with zero resistance[1]. However, there exists a critical current, $I_c$, above which the material cannot hold the SC state and a finite resistance appears. Understanding the mechanism of SC breakdown under a DC current is important for various SC applications. SC breakdown needs to be avoided, i.e., a large value of $I_c$ is required, when the SC material is used for carrying huge currents such as in the cables for SC magnets[2,3] and SC radio-frequency resonant cavities for particle accelerators[4,5]. On the other hand, SC breakdown upon irradiation of the material with visible, infrared, or terahertz photons under a DC bias offers an opportunity to utilize SCs as sensitive photon detectors[6–8].

Recently, a new functionality of a special group of SC materials has attracted much attention; the SC diode effect (SDE). Here, in principle, the DC resistance of a material in the SC state should be zero.

Meanwhile, a diode has different resistances depending on the polarity of the current (+ or − current). Therefore, to realize the diode effect in the SC state, the SC state needs to break down under + current but be held under − current, for example. In other words, the SDE is based on the nonreciprocity in the current-induced SC breakdown. In general, a material with broken inversion symmetry can host nonreciprocal transport[9,10], and it has been suggested that such a nonreciprocal current can be enhanced by the fluctuations of electrons in the SC state[11,12]. Among the variety of materials that show the SDE[11,13–19], artificial noncentrosymmetric superlattices made of stacked SC elements, in combination with the external magnetic field[20] or adjacent magnetic elements[21,22], were demonstrated to show a clear SDE that enables binary SC/normal-state switching, owing to the steepness of their SC transitions.

While the symmetry argument nicely explains whether non-reciprocal current is allowed, the microscopic mechanism

[1]Institute for Chemical Research, Kyoto University, Uji, Kyoto 611-0011, Japan. ✉e-mail: sekiguchi@crc.u-tokyo.ac.jp; kanemitu@scl.kyoto-u.ac.jp

underlying the SDE is still under intense discussion. Several theoretical studies have attributed the SDE to the SC-gap closing by the current injection, which occurs asymmetrically with respect to the current polarity. This behavior can be accounted for, for example, by assuming a Rashba-type electronic structure distorted by an in-plane magnetic field[23-26]. Here, the terahertz (THz) spectroscopy would be useful for experimental verification, because it enables us to directly monitor the SC gap, which lies on the meV energy scale. Furthermore, recent studies have demonstrated that irradiating an SC material with intense THz pulses triggers exotic nonlinear light-SC interactions, resulting in novel collective oscillations and efficient harmonic generations[27-31]. Therefore, THz spectroscopy of SDE materials would be an intriguing subject. However, the SC state inevitably becomes unstable under a large DC current close to $I_c$ where the SDE appears. To date, there is no report on how such an unstable SC state reacts to THz excitations. In general, excitations of Cooper pairs to form quasiparticles lead to a rapid transition to the normal state if the electric field of the THz pulses ($E_{THz}$) is strong enough[32], which can occur easily in the small-gap SC state. Or, it might be possible that a nontrivial response emerges from the THz-driven dynamics of an unstable system.

In this study, we investigate the interplay between the DC current and THz pulse excitations in a superconducting artificial superlattice. Our experimental setup allowed us to perform transport and spectroscopic experiments simultaneously. We found that THz pulse excitations affect the critical current in different ways depending on the THz frequency. Not only is the critical current sensitively reduced, but a peculiar SC/normal-state switching emerges under the DC and THz-pulse-induced

current, which can be ascribed to the vortex dynamics driven by the THz currents.

## Results

### Experimental setup

Figure 1 shows a schematic picture of the experimental setup. The [Nb (2.0 nm)/V (2.0 nm)/Ta (2.0 nm)]$_5$ superconducting artificial superlattice was epitaxially grown on a MgO (100) substrate with well-defined periodic interfaces. DC currents were injected through Ti/Au metal electrodes, which were deposited on both ends of the sample surface. These electrodes were also used to perform two-terminal current-voltage ($I$-$V$) measurements to monitor the SC/normal-state transition. The sample was cooled down in a magneto-optical superconducting magnet system. THz pulses were generated by using the tilted-pulse-front method with LiNbO$_3$[33] and were focused onto the sample after an intensity attenuation. THz spectroscopy was performed on a wide-film sample with a 4 mm-square area by using relatively broadband THz probe pulses with a weak peak-field strength of $E_{THz}$ = 5 kV/cm. On the other hand, for detailed measurements of the critical current under THz pulse excitations, another sample was fabricated into a wire structure, and narrowband (monochromatic) THz pulses were used [for details, see Supplemental Information].

### SC gap energy of artificial superlattice

Figure 2a shows the temperature dependence of the resistance measured at zero magnetic field. The sample shows a sharp transition to the SC state at $T_c$ = 4.0 K. Figure 2b compares the real parts of the THz conductivity spectra at 8.0 K and 1.6 K, measured at zero DC current and zero magnetic field. While the spectrum at 8.0 K is featureless, the conductivity in the low-energy region is suppressed at 1.6 K, reflecting the formation of the SC gap. The SC gap energy Δ at 1.6 K was determined to be 0.6 meV by fitting the spectrum with the Mattis-Bardeen model [see Supplemental Information], giving a spectral gap in the THz conductivity at 2Δ/$h$ of about 0.3 THz.

Next, we checked the SDE in the condition without THz pulse irradiations. An in-plane magnetic field of 50 mT was applied perpendicular to the DC current direction. Figure 2d plots the $I$-$V$ curve, where the jumps indicate that $I_c$ lies around 600 mA without THz irradiation. The finite gradient of the $I$-$V$ curve below $I_c$ comes from the resistance of the lead wires. Here, $I_c$ takes different values depending on the polarity relation between the magnetic field and DC current, demonstrating that nonreciprocity in the critical current indeed appears. Hence, let us move on to investigating the change in the SC gap depending on the DC current, $I_{DC}$. As shown in Fig. 2c, the conductivity spectrum around the SC gap hardly changes from $I_{DC}$ = 0 mA to 345 mA, pointing to the necessity of performing THz spectroscopy at a larger $I_{DC}$ close to $I_c$.

However, a further increase in $I_{DC}$ under the THz irradiation leads to the breakdown of the SC state. In other words, it turned out that the

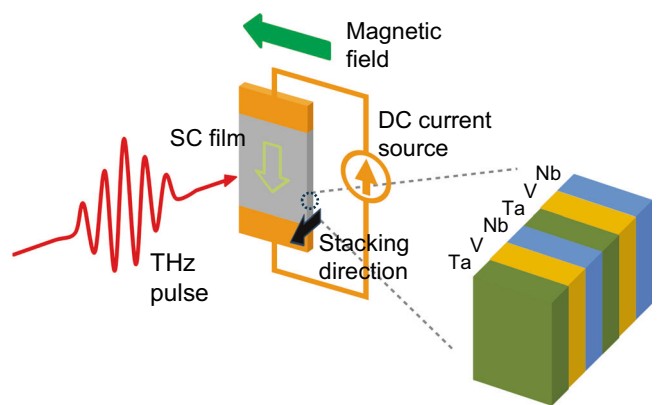

**Fig. 1 | Schematic drawing of the experimental setup.** The SC film sample is irradiated with THz pulses, while DC current flows through the film.

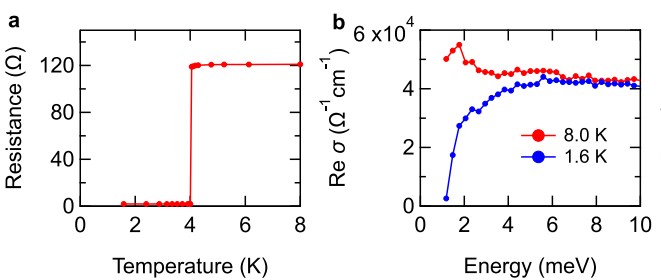

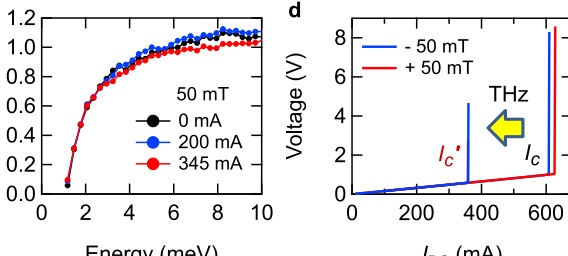

**Fig. 2 | Terahertz spectroscopy of the superconducting artificial superlattice [Nb (2.0 nm)/V (2.0 nm)/ Ta (2.0 nm)]$_5$. a** Temperature dependence of the resistance of [Nb(2.0 nm)/V(2.0 nm)/Ta(2.0 nm)]$_5$ film. **b** Real part of the THz conductivity spectra, measured at 8.0 K and 1.6 K. **c** Real part of the THz conductivity spectra, measured at 1.6 K, 50 mT, and different $I_{DC}$. The spectra are normalized by the normal-state spectrum measured at 8.0 K. **d** $I$-$V$ curves under in-plane magnetic field of ± 50 mT, with and without THz irradiation.

THz excitation significantly reduced the magnitude of critical current, as shown in Fig. 2d. In the experiment of the wide-area sample using a truly DC current, once the SC state is broken and switches to the resistive state, a large and continuous Joule heat is produced which compels us to shut off the DC current and halt the measurement. Therefore, THz spectroscopy cannot be performed at $I_{DC}$ larger than the THz-excitation-modified critical current, $I_c'$. One possible reason for the critical-current reduction is the increased sample temperature due to the average heating by the THz excitation. We excluded this trivial scenario by confirming that the sample temperature hardly changes upon the THz irradiation, through a measurement of the critical magnetic field [see Supplemental Information]. Another scenario to be considered is that the SC breakdown is solely triggered by the THz pulses, as reported elsewhere[32]. While we used a relatively weak-$E_{THz}$ probe for the spectroscopy, the THz pulse itself is capable of suppressing the SC gap through photoinjection of quasiparticles. However, we confirmed that the reduction in critical current occurs much more sensitively under the DC current, and shows different $E_{THz}$ dependence, compared to the nonlinearity induced solely by the THz pulses. This feature points to a crucial role of interplay between the THz excitation and DC current [for more details, see Supplemental Information]. In addition, the SDE cannot be discerned in the critical current reduced by the THz excitation. These observations imply the existence of nontrivial THz-induced dynamics that assist the DC-current-induced SC breakdown process.

## Modification of critical current by narrowband THz pulse excitations

To reveal the mechanism of the critical-current reduction induced by THz pulse excitations, we performed a detailed measurement of $I_c'$ using a different setup. The THz pulses were spectrally narrowed by band-pass filters to investigate the dependence on the THz frequency. Further, the film sample was fabricated into a wire structure with a 50-μm width and 900-μm length. Owing to the thin-wire structure, the critical current density can be reached using a smaller $I_{DC}$ (i.e., the magnitude of $I_c'$ becomes smaller), which facilitates the $I_c'$ measurement. The small dimensions of the sample also result in more uniform

spatial distributions of the DC current and THz field strength. In addition, we did not apply an external magnetic field in order to focus on the interplay between the DC current and THz pulse excitation, while there is a finite residual magnetic field in the cryostat.

Figure 3a–d show how the critical current changes depending on the $E_{THz}$ strength of the THz pulses for different THz frequencies. The available $E_{THz}$ strength at each frequency is determined by the spectrum of the broadband THz pulse [see Supplemental Information]. First, we discuss the cases where the THz photon energies are higher than the SC gap in the THz conductivity, $\nu = 0.8$ THz and 2.0 THz. As plotted in Fig. 3a, b, while $I_c'$ is not affected in the weakest $E_{THz}$ region, it is substantially reduced when $E_{THz}$ exceeds 3-5 kV/cm. Notably, this effect does not depend on the polarization of $E_{THz}$; the reduction in $I_c'$ is of the same magnitude regardless of whether the $E_{THz}$ is applied parallel or perpendicular to the DC current. This feature can be explained by considering the THz excitation process as a standard photon absorption; THz excitations with energies $> 2\Delta$ lead to quasiparticle excitations, whose efficiency is insensitive to the polarization of the $E_{THz}$. When a DC supercurrent is flowing, the Cooper pairs have finite momenta along the current direction; hence, their excitation produces quasiparticles with finite velocities and immediately leads to the appearance of a normal current component. The resultant Joule heat becomes larger when the $E_{THz}$ is stronger or the $I_{DC}$ is larger. When the amount of Joule heating exceeds the amount of heat dissipated to the environment, avalanche-type heating occurs which destroys the SC state.

On the other hand, different behavior is observed in the case of low-frequency THz excitations; $\nu = 0.2$ THz $< 2\Delta/h$ and 0.3 THz ~ $2\Delta/h$. Comparison between Fig. 3c, d and Fig. 3a, b shows that the reduction in $I_c'$ occurs at much lower field strengths around $E_{THz} = 0.3$–0.5 kV/cm at low-frequency excitations. More importantly, $I_c'$ is sensitive to the $E_{THz}$ polarization; the reduction in $I_c'$ is significantly larger when $E_{THz}$ is parallel to the DC current than when it is perpendicular. These features indicate that the low-frequency THz excitations assist the current-induced SC breakdown with a mechanism different from the standard quasiparticle excitation. This mechanism is further investigated in the following. When the THz excitation is below $2\Delta$, the THz irradiation

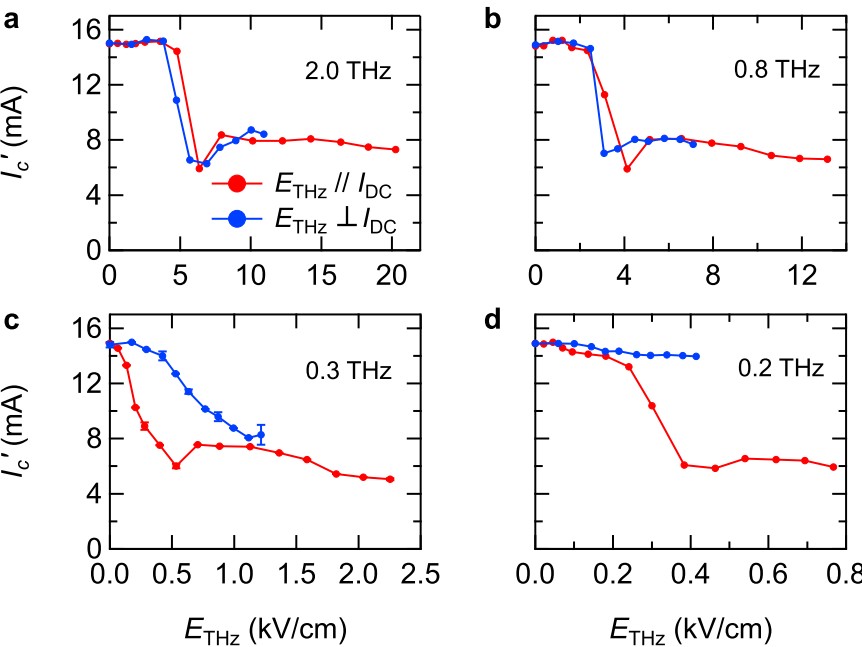

**Fig. 3 | $E_{THz}$ dependence of the critical current $I_c'$ at different THz frequencies.** **a, b** Results measured at 2.0 THz and 0.8 THz, which are higher energies than $2\Delta$. **c, d** Results measured at 0.3 THz and 0.2 THz, which are comparable and lower than $2\Delta$. Red (blue) points are results where the $E_{THz}$ polarization is parallel (perpendicular) to the DC current direction. Error bars in **c** are the standard deviation.

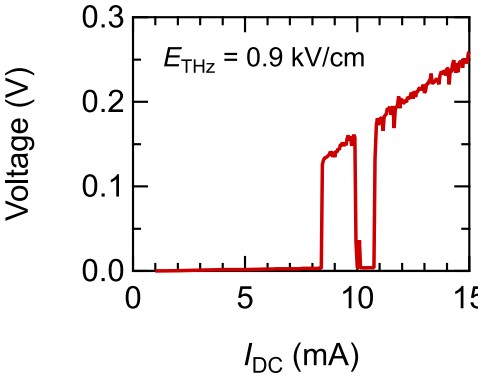

**Fig. 4 | *I-V* curve under the THz excitation, measured using the pulsed DC current.** THz frequency is 0.3 THz and $E_{THz}$ = 0.9 kV/cm.

does not induce quasiparticle excitations, but rather a coherent THz-oscillating current. This oscillating current is superposed onto the DC current in the sample, which boosts the SC breakdown process. Thus, how does the DC + THz current lead to the SC breakdown? A naïve criterion is that the SC state breaks down if the amplitude of the DC + THz current exceeds the critical current $I_c$. However, as described below, anomalous behavior incompatible with this criterion appears in the *I-V* curve.

Figure 4 shows the result of an *I-V* measurement with the THz excitation. The frequency is 0.3 THz, and $E_{THz}$ = 0.9 kV/cm is parallel to the DC current. Here, to investigate the $I_{DC}$ region above $I_c'$ while avoiding excess Joule heating, we used pulsed DC currents with a tiny duty ratio (a duration of 200 μs and a frequency of 50 Hz) in synchronization with the THz pulse excitation [for the detail of DC current pulse, see Supplemental Information]. In Fig. 4, as $I_{DC}$ is increased from zero, the voltage jumps to a high value at $I_c'$ = 8.4 mA, which is much smaller than the unmodified critical current $I_c$ = 15 mA. Notably, highly nontrivial behavior appears as $I_{DC}$ is further increased; a low resistive state appears again at an $I_{DC}$ above $I_c'$. As the resistance of the sample almost vanishes, this behavior indicates that the sample re-enters the SC state [for more details, see Supplemental Information]. Such non-monotonic SC/normal-state switching cannot be explained by the simple criterion that compares the amplitude of the DC + THz current with a particular threshold current, and it indicates that complex THz-induced dynamics underlie the SC breakdown. In the following, we show that a model based on the magnetic vortex dynamics driven by DC + THz current can reproduce the non-monotonic SC breakdown.

### THz-induced vortex dynamics in a trap potential

To proceed further, we consider what is the fundamental process that underlies the current-induced SC breakdown. Regarding the mechanism of current-induced SC breakdown in materials, two critical currents are widely known: the depairing current and the depinning current[1]. The depairing determines the intrinsic upper limit of the supercurrent in the material. As the current density increases, the SC gap energy shrinks due to the momentum of the Cooper pairs, leading to a decrease in the condensation density of the SC state. As a result, there exists a maximum supercurrent that can flow in the material, i.e., the depairing current, above which the SC state cannot be maintained[1,34,35]. On the other hand, the SC state can be destroyed before the depairing current is reached, by the Joule heat produced by vortex motions. In type-II SC materials, magnetic vortices penetrate the sample under a small magnetic field. These vortices are usually pinned to the trap potentials in the sample, which prevents the vortices from drifting. Therefore, the vortices do not contribute to transport phenomena as long as they are pinned. However, the DC current exerts Lorentz force on the vortex that effectively tilts the trap potential of the pinning center. When the depth of the potential

minimum becomes zero at a large $I_{DC}$, the vortex escapes from the potential and starts to drift, i.e., depinning of the vortex occurs. The drift of the vortex induces an electric field along the DC current resulting in the Joule heating, and destroys the SC state at a sufficiently large DC current density.

The residual magnetic field in the cryostat, which is on the scale of ∼10 mT, is enough to induce magnetic vortices in the SC film sample[36–38]. In addition, while the direction of the external magnetic field is nominally in the plane of the SC film, a tiny misalignment of the sample angle can easily introduce vortices perpendicular to the film[38]. Indeed, an out-of-plane magnetic field more sensitively suppresses the SC state than an in-plane one does [see Supplemental Information]. Hence, in the simulation described below, we assumed that the vortex is perpendicular to the film (while we note that it is possible to obtain similar results by assuming an in-plane vortex, if we modify the parameters in the model). If the vortex is depinned after the excitation by THz-induced oscillating current superimposed on the DC current, the vortex starts a continuous drift motion. The drift motion continues as long as the DC current exists, resulting in a sizable amount of Joule heat. On the other hand, if the vortex stays pinned after the THz pulse excitation, the vortex-induced Joule heating ceases within a few ps, which does not destroy the SC state. Therefore, in the following, we assume that the SC breakdown occurs if the vortex is depinned. The purpose of the simulation is to understand how the vortex depinning is affected by the THz-induced current.

The equation of motion of the vortex along the direction perpendicular to the DC and THz current is given by[30,39]

$$m\frac{dv}{dt} = -\left[J_{DC} + J_{THz}\right]d_{film}\Phi_0 - \frac{dV_{trap}}{dy} - \eta d_{film}v \quad (1)$$

where $m$, $v$, and $y$ are the mass, velocity, and position of the vortex, respectively, $J_{DC}$ and $J_{THz}$ are the DC and THz current densities, $\Phi_0$ is the flux quantum, and $d_{film}$ = 30 nm is the film thickness. The first term in Eq. (1) represents the Lorentz force from the currents, while the second term is the restoring force by the trap potential $V_{trap}$, and the third term is the viscous drag with a coefficient $\eta$. We assume that $V_{trap}$ has a Gaussian shape. Considering that the local minimum of the $V_{trap}$ disappears at the critical current density $J_c$[40], $V_{trap}$ can be expressed as

$$V_{trap} = -d_{film}\Phi_0 J_c\sqrt{\frac{e}{\alpha}}\exp\left[-\frac{\alpha}{2}y^2\right] \quad (2)$$

where $\alpha$ represents the inverse potential width. In the following, we use the current amplitude instead of the current density for an easy comparison with the experimental data. The THz-induced current is given by

$$I_{THz}(t) = I_{THz}\sin(-\omega t)\exp\left[-\frac{t^2}{p}\right] \quad (3)$$

where the frequency $\omega/2\pi$ and the pulse duration $2\sqrt{p}$ correspond to the experimental values, 0.3 THz and 23 ps. The irradiation with peak $E_{THz}$ = 1 kV/cm leads to the $I_{THz}$ with a peak amplitude of 24 mA, as calculated from the THz conductivity of the SC state [for details, see Supplemental Information]. Taking into account this $E_{THz}$-$I_{THz}$ correspondence and the experimental $E_{THz}$ dependence shown in Fig. 3c, the region of interest for the simulation is $I_{THz}$ = 0-40 mA. We assume that the critical current without the THz current is $I_c$ = 15 mA.

We simulate the vortex dynamics to see qualitatively how the vortex depinning changes depending on the DC and THz current. We adopt the value of $\alpha$ so that the simulation reproduces the $E_{THz}$ dependence in Fig. 3, given a mass per unit length of the vortex of $10^7 m_e$/m and damping constant of 1 THz[30,41]. 

Figure 5a plots the effective trap potential as a function of the vortex position $y$, where we have assumed $\alpha$ = 2.5 μm$^{-2}$. Under the DC current, the trap potential is tilted by the Lorentz force, while at $I_{DC}$ = 11 mA < $I_c$ the vortex is still trapped at the potential minimum

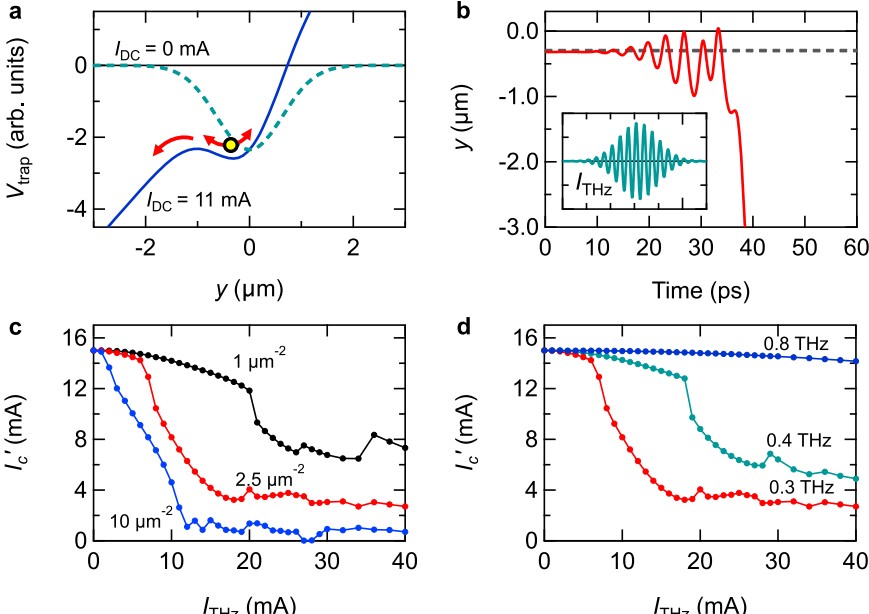

**Fig. 5 | Simulation of the THz-current-assited vortex depinning. a** Trap potential for the vortex, at $I_{DC} = 0$ mA and 11 mA. **b** Simulated vortex motion driven by the DC + THz currents. The inset shows the waveform of $I_{THz}$, whose frequency is 0.3 THz and peak amplitude is 8 mA. $I_{DC}$ is 11 mA. **c** $I_{THz}$ dependence of the critical current, calculated with different values of the potential width $\alpha$. The frequency is 0.3 THz. **d** $I_{THz}$ dependence of the critical current calculated for different THz frequencies. Here, $\alpha$ is 2.5 $\mu m^{-2}$.

around $y = -0.3$ μm. On top of the DC current, the THz current exerts Lorentz force on the vortex inducing an oscillation, and the vortex escapes from the trap potential when the amplitude of the oscillation becomes large enough. Figure 5b shows an example of the simulated vortex motion. The peak amplitude of the THz current is set to $I_{THz} = 8$ mA, which is almost half of $I_c$. Once the oscillating motion grows large enough, the vortex escapes from the potential and starts to drift; i.e., the vortex is depinned by the THz current. The drift motion persists as long as the DC current exists, leading to continued Joule heating that destroys the SC state. As shown in Fig. 5b, the simulation shows that the vortex can be deppined at an $I_{DC}$ smaller than $I_c$ with the assistance of the THz current. Therefore, we can define $I_c'$ to be the minimum $I_{DC}$ value at which depinning occurs under a particular THz excitation condition.

Figure 5c plots the simulated value of $I_c'$ as a function of $I_{THz}$ for different potential widths $\alpha$. When $\alpha$ is small, i.e., the potential is wide, a larger THz-induced oscillation is necessary for depinning, and hence, $I_c'$ is insensitive to $I_{THz}$. Among the curves, the one for $\alpha = 2.5$ $\mu m^{-2}$ starts to decrease rapidly around $I_{THz} = 10$ mA and converges to a value of about 4 mA in the higher $I_{THz}$ region. This behavior well reproduces the experimental results in Fig. 3c, taking into account the correspondence that the THz current of $I_{THz} = 10$ mA is induced by THz pulses with $E_{THz} = 0.4$ kV/cm. Hence, we use $\alpha = 2.5$ $\mu m^{-2}$ in the following simulations (while we note that a particular choice is not crucial for a qualitative discussion on THz-current-assisted depinning). Figure 5d shows the reduction in $I_c'$ for different THz frequencies. Unlike the 0.3-THz case, $I_c'$ is less sensitive to $I_{THz}$ in the 0.4-THz case and it hardly changes in the 0.8-THz case, indicating the small impact of THz excitations at higher frequencies. This is because the amplitude of the vortex oscillation becomes smaller for a higher frequency force [see Supplemental Information for the influence of the pulse duration]. Therefore, the vortex dynamics are less affected by higher-frequency THz excitations, which is consistent with the experimental results in Fig. 3 where the $I_c'$ reduction by the THz excitation is less effective at higher frequencies.

The simulated vortex behavior described above supports that the vortex depinning model successfully explains the experimental results

of SC breakdown. We should note that, same as in the experiment, the nonreciprocity in $I_c'$ cannot be simulated based on this model. This may be due to the dominant role of the THz-current-driven vortex dynamics in determining $I_c'$ that obscures the delicate asymmetry of the trap potential [see Supplemental Information].

Given the validity of the THz-current-assisted vortex depinning model, we now simulate the behavior shown in Fig. 4, i.e., how the vortex dynamics change with increasing $I_{DC}$ under the $I_{THz}$ of a particular amplitude. The peak amplitude of $I_{THz}$ is set to 8 mA, the same value as in Fig. 5b. The solid lines in Fig. 6a represent the vortex motion at different $I_{DC}$ in the time window around the peak of $I_{THz}$ [for the full-time window, see Supplemental Information]. The dashed line shows the THz-current-induced Lorentz force, $F_{THz}$, exerted on the vortex.

At a relatively small $I_{DC}$ of 9 mA where the minimum of the trap potential is relatively deep, the vortex does not escape from the potential. On the other hand, at $I_{DC} = 11$ mA, the potential minimum becomes shallower, allowing the vortex to be depinned (hence, $I_c'$ lies between 9 mA and 11 mA). Notably, a further increase in $I_{DC}$ triggers peculiar vortex dynamics. At $I_{DC} = 11.7$ mA, the vortex does not escape from and stays in the potential, in spite of the even shallower potential minimum. This result indicates that it is possible to pin the vortex under a DC current larger than $I_c'$. At $I_{DC} = 12.1$ mA, the vortex is again depinned during the oscillation. This non-monotonic vortex depinning corresponds to the non-monotonic SC/normal-state switching in Fig. 4.

Figure 6c shows whether the vortex stays pinned or gets depinned depending on $I_{DC}$. The vortex can be "repinned" by the potential above $I_c'$, where the SC state can be maintained. In particular, there is a wide repinning region of 11.41 mA < $I_{DC}$ < 11.96 mA, whereas there is another spike-like region around 12.26 mA. While the inhomogeneity of the trap potential or the fluctuation of the $E_{THz}$ strength can obscure the spike-like repinning region, the wide repinning region can be robust against such fluctuations, allowing the experimental observation of repinning. We note that the $I_{DC}$ region where vortex repinning occurs is sensitive to the amplitude of $I_{THz}$. The wide repinning region does not appear at the larger values of $I_{THz}$, which is consistent with the experimental results [see Supplemental Information].

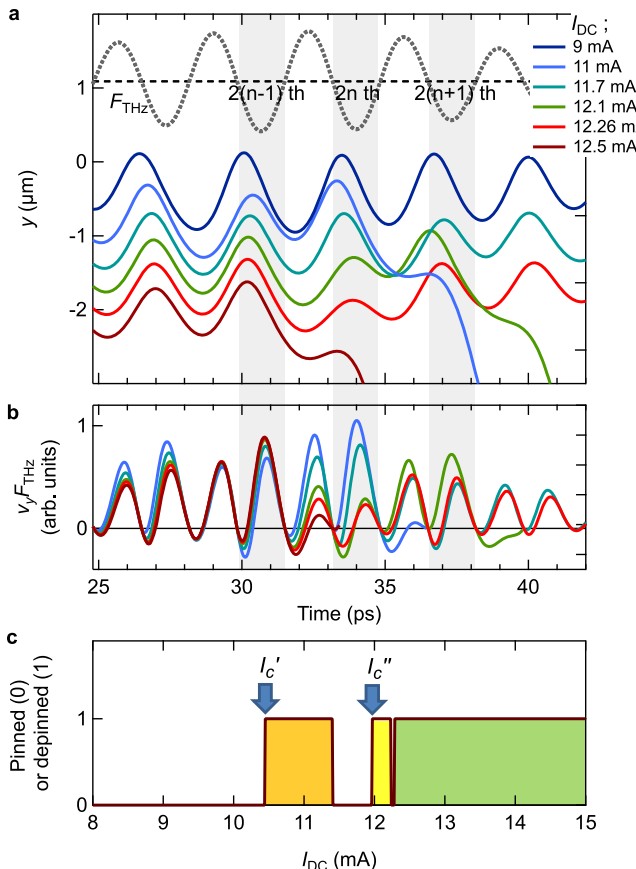

**Fig. 6 | Simulation of the THz-current-driven vortex dynamics. a** Vortex motion at different $I_{DC}$ in the time window around the peak of the THz pulse. The frequency is 0.3 THz and the peak amplitude of $I_{THz}$ is 8 mA. The dashed line shows the Lorentz force exerted on the vortex. **b** Power exerted on the vortex at different $I_{DC}$. For an easy view, the curve at the lowest DC current, $I_{DC} = 9$ mA, is not shown. In addition, we do not plot the power after the vortex is deppined. **c** Plot indicating whether the vortex is pinned or depinned depending on $I_{DC}$, under the influence of $I_{THz}$. The depinning region are filled with different colors, depending on at which cycle of the THz pulse the depinning occurs.

## Discussion

Now we discuss what factor the non-monotonic $I_{DC}$ dependence of the vortex depinning stems from. As can be seen in Fig. 6a, the vortex can be depinned at different cycles of oscillation depending on $I_{DC}$. Note that, due to the polarity of the DC current, the vortex can be depinned and launched only in one direction. Hence, we assume that the vortex is depinned at the $2n$-th cycle of the $F_{THz}$ oscillation. We sort the depinning region in Fig. 6c by $n$, i.e., at which cycle the vortex is depinned. This sorting reveals that the vortex repinning appears as a gap region between different depinning dynamics. This feature can already be observed in Fig. 6a; below and above the first repinning region ($I_{DC} = 11$ mA and 12.1 mA, for example), the vortex is depinned at the $2n$-th and $2(n+1)$-th cycle of the $F_{THz}$ pulse, respectively. Similarly, below and above the second repinning region ($I_{DC} = 12.1$ mA and 12.5 mA, for example), the vortex is depinned at the $2(n+1)$-th and $2(n-1)$-th cycle, respectively.

The emergence of different vortex depinning dynamics can be understood as follows; as $I_{DC}$ is increased from zero, the trap potential continuously becomes shallower. When $I_{DC}$ reaches $I_c'$, the vortex is able to escape from the potential at the $2n$-th cycle of $F_{THz}$, which is around the pulse peak. As $I_{DC}$ is increased further, the potential becomes shallower, and eventually at $I_{DC} = I_c''$, the vortex can escape from the potential at another time, the $2(n+1)$-th cycle of the

oscillation. On the other, the vortex depinning at the $2n$-th cycle does not occur at $I_{DC} = I_c''$. This is due to the phase mismatch between $F_{THz}$ and the vortex oscillation, which leads to the suppression of the oscillation amplitude of the vortex. When the potential width becomes shallow, the vortex is driven around the edge region of the trap potential. There, the restoring force from the potential becomes tiny due to the flat curvature of the potential, inducing a large phase shift in the vortex oscillation relative to $F_{THz}$. In other words, the anharmonicity of the potential endows the vortex motion with a complex phase shift depending on $I_{DC}$.

In order to discuss the phase relation between the vortex motion and $F_{THz}$, we plot the power exerted on the vortex, $v_y F_{THz}$, in Fig. 6b. The vortex motion and $F_{THz}$ are "in phase" when they move simultaneously in the negative direction. This results in a large value of $v_y F_{THz}$. In Fig. 6b, $v_y F_{THz}$ at the $2n$-th cycle is large at $I_{DC} = 11$ mA, enabling the vortex depinning there. However, a further increase in $I_{DC}$ results in a smaller $v_y F_{THz}$ at the $2n$-th cycle, indicating that the in-phase condition is diminished. As a consequence, a vortex repinning appears above $I_c'$. Similar behavior can be discerned in the depinning at other cycles. Therefore, Fig. 6a-c tell us that the phase of the vortex motion in the anharmonic trap potential plays a crucial role in the depinning of vortices, leading to the peculiar DC-current dependence of SC breakdown and recovery.

In summary, we investigated the current-induced SC breakdown process in a superconducting artificial superlattice irradiated by THz pulses. We found that the critical current is modified by the THz pulse excitations through different mechanisms depending on the THz frequency. Above-gap high-frequency excitations induce quasiparticle excitations, leading to a polarization-insensitive reduction in the critical current. On the other hand, below-gap low-frequency excitations trigger a polarization-sensitive reduction in the critical current at small $E_{THz}$ strengths. Not only is the critical current sensitively reduced, but anomalous behavior appears that the supercurrent flows above the modified critical current. By considering vortex depinning as the origin of current-induced SC breakdown, we showed that the simulated vortex dynamics under DC and THz-driven current successfully reproduce the features observed at low-frequency THz excitations. While the interplay of the DC current and THz excitation hinders observation of the delicate nonreciprocity in the critical current, it endows SC materials with novel functionalities. The sensitivity of the SC breakdown to the $E_{THz}$ polarization can lead to a new THz and microwave photodetector that discriminates the polarization direction of those low-energy photons. A SC material with non-monotonic SC/normal-state switching might work as a new component that shows distinct performance from the existing devices in future quantum electronic applications, for example, in non-volatile memories and logic circuits with ultralow power consumption. Therefore, the results obtained in this study will provide an alternative insight into the development of SC devices based on the SC breakdown process.

## Methods

A detailed explanation of the experimental setup is provided in S1 in Supplemental Information. The [Nb (2.0 nm)/V (2.0 nm)/Ta (2.0 nm)]$_5$ superconducting artificial superlattice was epitaxially grown on a MgO (100) substrate at 700 °C by direct current magnetron sputtering in a high-vacuum system with a base pressure of ~5 × 10$^{-6}$ Pa. DC currents were injected into the superlattice through Ti/Au metal electrodes deposited on both ends of the film sample. The DC current injection and the two-terminal current-voltage ($I$-$V$) measurement were performed using a DC voltage current source/monitor. THz pulses were generated by the tilted-pulse-front method with a LiNbO$_3$. After an intensity attenuation using a pair of wire-grid polarizers, the THz pulses were focused onto the sample. THz spectroscopy was performed in the transmission geometry. $E_{THz}$ was either parallel or perpendicular to the DC current. The sample was cooled down in a magneto-optical

cryogen-free superconducting magnet system. The magnetic field was applied in the plane of the film and perpendicular to the DC current. For the THz spectroscopy, a wide-area film sample (4-mm square) was used. On the other hand, for the detailed investigation of $I_c'$, the film sample was patterned into a 50-μm-wide and 900-μm-long wire structure by using a conventional photolithography and Ar ion milling process. Optical images of the samples are shown in Fig. S1–1b, c.

## Reporting summary

Further information on research design is available in the Nature Portfolio Reporting Summary linked to this article.

## Data availability

All data needed to evaluate the conclusions in the paper are present in the paper and/or the Supplementary Information. Additional data and code related to this paper are available from the corresponding author upon request.

## Code availability

All data needed to evaluate the conclusions in the paper are present in the paper and/or the Supplementary Information. Additional data and code related to this paper are available from the corresponding author upon request.

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

## Acknowledgements

Part of this study was supported by a grant from the Japan Society for the Promotion of Science (JSPS; KAKENHI Grant No. JP19H05465 (Y. K.), No. JP21K18145 (T. O.), No. JP21K13883 (H. N.), No. JP23K13045 (F. S.)), The

FUTABA Foundation (H. N.), Iketani Science and Technology Foundation (H. N.), Sumitomo Foundation (H. N.), and the Collaborative Research Program of the Institute for Chemical Research, Kyoto University (F. S. and H. N.).

## Author contributions

F.S. performed the experiments and simulations with the help of H.N. and H.H. H.N. fabricated the samples. All authors contributed to the interpretation of the results. F.S. wrote the paper with input from all authors. F.S. and H.H. conceived, and Y.K. and T.O. supervised the project.

## Competing interests

The authors declare no competing interests.
