## [Peer Review File · Nature Communications]

Anomalous behavior of critical current in a superconducting film triggered by DC plus terahertz currentREVIEWER COMMENTS

Reviewer #1 (Remarks to the Author):

Dear editor,

working through the manuscript "Anomalous behavior of critical current in a superconducting film triggered by DC plus terahertz current" by F. Sekiguchi et al. I do find that the work is interesting but I feel 1) the data is not comprehensive and 2) the data does not unambiguously is understood with the model picture given. The presented model of the vortex dynamics could be the one possible explanation for the data but I do not see how the data points to this as the unique picture.

Additionally I do find the paper hard to follow. While in some basic parts the authors give nearly textbook knowledge explanations, I do feel it jumpy and handwaving on less obvious parts. Also some technical details are missing. In particular the characterization of the absolute field strength of the THz pulses or the details on how they estimate the current strengths.

The introduction is fine but a bit lengthy. I do miss an explanation on the sample design, why this specific design is chosen to achieve an SDE. Also I would show the stacking direction already in the main text. What is confusing is that the text speaks about contacts to the surface but the sketch in S1 shows contacts to the edges.

The characterization of the sample is fine. However I wonder if its not instructive throughout the paper to also discuss the imaginary part of the conductivity. If the authors are right with the vortex picture that should show up far more sensitive in the phase response and thus the imaginary $1/\omega$ response of the superconductor. Since they measure time resolved spectra the field changes might also be more sensitive on a phase shift of the pulse. The latter I suspect also to be a far more sensitive probe of possible heating effects rather than using critical magnetic fields as a reference.

I can follow the description up to the point where they authors increase the current up to 345mW (Fig 2d). The part of the break down of SC at higher current I feel missing a lot. Like how does the spectrum look like right at reaching the critical current? How does that scale with changing the THz fluence?

Maybe a fluence dependence of the spectra σ_1 and σ_2 (and/or amplitude and phase of the time domain pulse) might be very instructive. Both at 345mA and above. Even a power dependent spectrum without current would be useful to characterize at what field strength non-linearities are induced by the THz pulse set in and how potentially vortices cut down the coherence length.

I agree that there is a clear difference between above and below gap THz excitation. I am surprised why exactly at gap pumping is not much more efficient in pair breaking as dominant effect.

Since the scales are all different it would be nice to see if within the errorbars the dependencies at 0.3THz and 0.2 THz are the same.

From now on the authors use the current dependence and the non linearities found there to investigate the mechanism.

I wonder if here a linear THz probe of the system with a weak broad band pulse during the narrowband excitation would be a suitable probe to see a spectral response to foster also the model the authors present.

This would also allow characterizing the re-entrant like feature a true SC state or some non-equilibrium state of highly mobile carriers.

For their model the authors claim induced vortices and their dynamics under the THz field as core reason for the observed features. Do the effects disappear if they redo the experiments in a truly non-magnetic cryostate?

As said the model the authors use can describe the experimental observation of Fig.4 but to show that the system indeed follows that can't the authors show a change of I_c' and I_c'' as function of THz frequency and field. S8 only shows an example with and without reentrance behaviour. A predicted shift of I_c'' would be much more powerful.

Also I do miss a clear discussion of the pulse length dependence. How can the vortex dynamics be so sensitive on changes between 2.5 and 23 ps while the current is basically measured on dc scales of the current pulses that I suspect are orders of magnitudes longer. Shouldn't the vortices relax back already for most of the measurement time?

In total I do find the presented data and the found features interesting but I do not see a clear story/model that is backed uniquely by the data. So I feel a more detailed characterization of the nonlinearity, ideally spectroscopically would strengthen the manuscript a lot and maybe even help to strengthen the vortex case of the model.

At the present stage I can not support publication of the present manuscript.

Reviewer #2 (Remarks to the Author):

Attached

Reviewer #3 (Remarks to the Author):

The authors investigate the impact of terahertz (THz) field excitations on the critical current of a superconductor (SC) in SC devices, particularly SC diodes. Through experimentation involving the application of both terahertz (THz) and direct current (DC) in a superconducting artificial superlattice, the authors observe a sensitive modification of the critical current, displaying a nontrivial relationship with THz excitations. The interplay between THz and DC current leads to intriguing non-monotonic SC/normal state switching. However, my ability to fully assess the validity of the findings is hindered by the absence of crucial information. To enhance the manuscript, I recommend addressing the following concerns:

1. AC Current Induced by THz Field Alone:

A preliminary discussion on whether the AC current induced by the THz field alone without DC field can break down the superconducting state is essential. Specifically, clarification is needed regarding the potential of peak AC current, comparable to the DC critical current, to break the superconducting state without applying any DC field. This foundational understanding is crucial before delving into the investigation of the combined AC + DC current scenario presented in the paper.

2. Comparison of Peak AC Current and DC Current:

The manuscript lacks clarity regarding how the peak AC current induced by THz pulses compares to the DC current. Providing this information is vital for a comprehensive assessment of the results. Additionally, it would be valuable to understand the breakdown of the superconducting state solely due to THz field-induced current. Clarify whether the THz field-induced current alone can disrupt the SC state and, if so, provide details on the peak current amplitude in comparison to the critical current.

3. Effect of THz Field on SC Diode:

The manuscript does not clearly address the impact, if any, of the THz field on the SC diode effect. It appears that there might be no effect or a reduction in the effect. Please provide clarification on whether the THz field influences the SC diode effect, and if so, elucidate the extent of its impact.

4. Inclusion of Error Bars in Figure 3:

Figure 3, representing the main results, requires the inclusion of error bars for data points. This

addition is crucial for a more accurate interpretation and validation of the presented findings.

In summary, addressing these points will strengthen the manuscript and enable a more thorough evaluation of the research. I cannot currently recommend publication in the present state but believe that incorporating these suggestions will significantly enhance the overall quality of the manuscript.

Reply to Reviewer #1

Report by Reviewer #1

I have reviewed the response and new manuscript of the authors and I am mostly satisfied with the answers. The storyline became much more clear and detailed at the important steps. The authors added a lot of additional information to substantiate their claims and revised the manuscript accordingly. Also the responses to all reviewers questions and comments are presented in a clear and honest way about what are the limits of the present experimental setup etc. Well done, from my point of view.

Reply

We thank Reviewer #1 for his/her careful reading of our reply and the revised manuscript. We are happy that the Reviewer is satisfied with our responses, and finds that the manuscript became much more clear.

Report by Reviewer #1

I do only have two minor questions/comments. Its still not clear too me how there authors do estimate the field strength. That should crucially depend on the four spot size. The formula given does not include any spot size just a $\Delta I / I$.

Reply

The field strength of the THz pulses (E_{THz}) can be estimated by the formula given in S1 in the Supplemental Information, where the E_{THz} is proportional to the signal from the balanced photodetector, $\Delta I / I$. The E_{THz} indeed becomes weaker when the spot size of the THz pulse becomes wider, while it does not affect the validity of the formula. In other words, the amplitude of the EO sampling signal $\Delta I / I$ is a direct measure of the E_{THz} . Hence, we can directly determine the E_{THz} strength at the EO sampling position.

On the other hand, the THz pulses have different spot sizes at the sample position (inside the cryostat) and at the EO sampling position, due to the different focal lengths of the parabolic mirrors. The spot size at the sample position is estimated to be larger than the EO sampling position, by a factor of 4. The difference in the spot size was taken into account when we estimated the E_{THz} at the sample position.

Change: In the explanation of how we estimated the THz field strength in “S1. Methods” in the Supplemental Information, we added a sentence “taking into account the larger spot size of THz pulses at the sample position, by a factor of 4, compared to the spot size at the EO sampling position.”

Report by Reviewer #1

The point of which the authors discuss the reentrance of potential SC. They still call it like this as if the state is proven. I would prefer that they call it a high mobility state and propose that it is a possible SC state based on their model assumptions. For a proof one would have to measure real

SC properties.

Reply

We thank the Reviewer for the comment. We agree with the Reviewer that it is “an assumption” that the high-mobility state is the SC state, while we think this assumption is reasonable. Following the Reviewer’s suggestion, we added a new section in the Supplemental Information where we discuss the attribution of the emergent low resistive state.

Change: On page 4, we added a sentence claiming that we assume the low resistive state is the SC state. In the Supplemental Information, we added a new section S17. Attribution of the low resistive state emergent above I_c .

Report by Reviewer #1

With hopefully correcting/clarifying these two little things I would be in support of publishing the present manuscript.

Reply

We greatly appreciate the Reviewer’s comments that helped us improve the manuscript.

Reply to Reviewer #2

Report by Reviewer #2

The authors have answered most of the questions I had in a satisfactory manner. They have significantly improved the presentation of the manuscript and added relevant technical details required for understanding the relevance of their results.

Reply

We thank Reviewer #2 for his/her careful reading of our reply and the revised manuscript. We are happy that the Reviewer is satisfied with our responses, and finds that the quality of the manuscript was improved.

Report by Reviewer #2

However, for one of the questions, they misunderstood what I asked for. This is regarding the THz excitation using different THz frequencies. They have shown the broadband THz pulse and the spectrum generated from the LNO crystal. This is fine. However, what I wanted to see are the narrowband THz pulses and the corresponding spectra that they generated using the metal-mesh filters. These are the actual input pulses that the authors used for exciting the sample. Could the authors provide all these pulses and the spectra, for 0.2 THz, 0.3 THz, 0.8 THz and 2 THz in the supplementary information. In addition, providing the relevant source (company or otherwise!) for such a filter would be beneficial for the community. I request the authors to make this revision after which I can provide my final decision.

Reply and change

The waveforms and spectra of the narrowband THz pulses, for 0.2 THz, 0.3 THz, 0.8 THz, and 2 THz, are now all shown in Fig. S6 in the Supplemental Information. The metal-mesh band-pass filters are from Origin Ltd (we are not sure the information about the company is available in English).

REVIEWERS' COMMENTS

Reviewer #1 (Remarks to the Author):

I have reviewed the response and new manuscript of the authors and I am mostly satisfied with the answers. The storyline became much more clear and detailed at the important steps. The authors added a lot of additional information to substantiate their claims and revised the manuscript accordingly. Also the responses to all reviewers questions and comments are presented in a clear and honest way about what are the limits of the present experimental setup etc. Well done, from my point of view.

I do only have two minor questions/comments. Its still not clear too me how there authors do estimate the field strength. That should crucially depend on the four spot size. The formula given does not include any spot size just a $\Delta I / I$. The point of which the authors discuss the reentrance of potential SC. They still call it like this as if the state is proven. I would prefer that they call it a high mobility state and propose that it is a possible SC state based on their model assumptions. For a proof one would have to measure real SC properties.

With hopefully correcting/clarifying these two little things I would be in support of publishing the present manuscript.

Reviewer #2 (Remarks to the Author):

The authors have answered most of the questions I had in a satisfactory manner. They have significantly improved the presentation of the manuscript and added relevant technical details required for understanding the relevance of their results. However, for one of the questions, they misunderstood what I asked for. This is regarding the THz excitation using different THz frequencies. They have shown the broadband THz pulse and the spectrum generated from the LNO crystal. This is fine. However, what I wanted to see are the narrowband THz pulses and the corresponding spectra that they generated using the metal-mesh filters. These are the actual input pulses that the authors used for exciting the sample. Could the authors provide all these pulses and the spectra, for 0.2 THz, 0.3 THz, 0.8 THz and 2 THz in the supplementary information. In addition, providing the relevant source (company or otherwise!) for such a filter would be beneficial for the community. I request the authors to make this revision after which I can provide my final decision.

Reply to Reviewer #1

Report by Reviewer #1

I have reviewed the response and new manuscript of the authors and I am mostly satisfied with the answers. The storyline became much more clear and detailed at the important steps. The authors added a lot of additional information to substantiate their claims and revised the manuscript accordingly. Also the responses to all reviewers questions and comments are presented in a clear and honest way about what are the limits of the present experimental setup etc. Well done, from my point of view.

Reply

We thank Reviewer #1 for his/her careful reading of our reply and the revised manuscript. We are happy that the Reviewer is satisfied with our responses, and finds that the manuscript became much more clear.

Report by Reviewer #1

I do only have two minor questions/comments. Its still not clear too me how there authors do estimate the field strength. That should crucially depend on the four spot size. The formula given does not include any spot size just a $\Delta I / I$.

Reply

The field strength of the THz pulses (E_{THz}) can be estimated by the formula given in S1 in the Supplemental Information, where the E_{THz} is proportional to the signal from the balanced photodetector, $\Delta I / I$. The E_{THz} indeed becomes weaker when the spot size of the THz pulse becomes wider, while it does not affect the validity of the formula. In other words, the amplitude of the EO sampling signal $\Delta I / I$ is a direct measure of the E_{THz} . Hence, we can directly determine the E_{THz} strength at the EO sampling position.

On the other hand, the THz pulses have different spot sizes at the sample position (inside the cryostat) and at the EO sampling position, due to the different focal lengths of the parabolic mirrors. The spot size at the sample position is estimated to be larger than the EO sampling position, by a factor of 4. The difference in the spot size was taken into account when we estimated the E_{THz} at the sample position.

Change: In the explanation of how we estimated the THz field strength in “S1. Methods” in the Supplemental Information, we added a sentence “taking into account the larger spot size of THz pulses at the sample position, by a factor of 4, compared to the spot size at the EO sampling position.”

Report by Reviewer #1

The point of which the authors discuss the reentrance of potential SC. They still call it like this as if the state is proven. I would prefer that they call it a high mobility state and propose that it is a possible SC state based on their model assumptions. For a proof one would have to measure real

SC properties.

Reply

We thank the Reviewer for the comment. We agree with the Reviewer that it is “an assumption” that the high-mobility state is the SC state, while we think this assumption is reasonable. Following the Reviewer’s suggestion, we added a new section in the Supplemental Information where we discuss the attribution of the emergent low resistive state.

Change: On page 4, we added a sentence claiming that we assume the low resistive state is the SC state. In the Supplemental Information, we added a new section S17. Attribution of the low resistive state emergent above I_c .

Report by Reviewer #1

With hopefully correcting/clarifying these two little things I would be in support of publishing the present manuscript.

Reply

We greatly appreciate the Reviewer’s comments that helped us improve the manuscript.

Reply to Reviewer #2

Report by Reviewer #2

The authors have answered most of the questions I had in a satisfactory manner. They have significantly improved the presentation of the manuscript and added relevant technical details required for understanding the relevance of their results.

Reply

We thank Reviewer #2 for his/her careful reading of our reply and the revised manuscript. We are happy that the Reviewer is satisfied with our responses, and finds that the quality of the manuscript was improved.

Report by Reviewer #2

However, for one of the questions, they misunderstood what I asked for. This is regarding the THz excitation using different THz frequencies. They have shown the broadband THz pulse and the spectrum generated from the LNO crystal. This is fine. However, what I wanted to see are the narrowband THz pulses and the corresponding spectra that they generated using the metal-mesh filters. These are the actual input pulses that the authors used for exciting the sample. Could the authors provide all these pulses and the spectra, for 0.2 THz, 0.3 THz, 0.8 THz and 2 THz in the supplementary information. In addition, providing the relevant source (company or otherwise!) for such a filter would be beneficial for the community. I request the authors to make this revision after which I can provide my final decision.

Reply and change

The waveforms and spectra of the narrowband THz pulses, for 0.2 THz, 0.3 THz, 0.8 THz, and 2 THz, are now all shown in Fig. S6 in the Supplemental Information. The metal-mesh band-pass filters are from Origin Ltd (we are not sure the information about the company is available in English).